# Efficacy of Propolis Gel on Mature Biofilm Formed by *Neocosmospora keratoplastica* Isolated from Onychomycosis

**DOI:** 10.3390/jof8111216

**Published:** 2022-11-17

**Authors:** Polyana de Souza Costa, Elton Cruz, Flávia Veiga, Isabelle Carrilho Jarros, Melyssa Negri, Terezinha Inez Estivalet Svidzinski

**Affiliations:** Medical Mycology Laboratory, Department of Clinical Analysis and Biomedicine, State University of Maringá, Maringá 87020-900, Brazil

**Keywords:** natural products, propolis extract, antifungal activity, fungal nail infection, ex vivo model

## Abstract

This article describes *Neocosmospora keratoplastica* as an etiological onychomycosis agent. Ex vivo studies were initially performed to demonstrate the ability of this species to grow and form a well-organized characteristic biofilm on sterilized healthy nails. Based on the history of excellent results, both for antifungal and antibiofilm, of propolis resin gum, we evaluated its activity using artificially formed biofilm. In vitro, the minimal biofilm eradication concentration of the propolis extract (PE) was 375 µg of total polyphenol content (TPC) per mL, while for the propolis gel (PG) it was 450 µg of TPC per mL. In biofilm exposed to the propolis products, a decrease in hyphae and conidia was evident, accompanied by a disorganization of the extracellular matrix. Additionally, this low concentration of PE was able to significantly reduce the number of colony-forming units and the metabolic activity. Furthermore, the treatment of a 15-year nail infection due to *N. keratoplastica* was carried out exclusively using a topical treatment with a gel containing propolis (30%) with a daily dosage. This treatment achieved complete remission of the onychomycosis in 12 months. It is important to point out that some inconveniences previously reported by other patients treated with propolis extract were eliminated, increasing adherence to treatment.

## 1. Introduction

Onychomycosis (OM) is a fungal nail infection which is responsible for approximately 50% of all human nail diseases. Its prevalence worldwide is up to 5.5% [1]. The rich symptomatology leads to discomfort and social repercussions, affecting quality of life [1,2,3]. Despite this, the etiopathogenesis of OM is still unclear, and there are many problems regarding its treatment. 

Most cases of OM are caused by dermatophytes, but the number of cases caused by non-dermatophyte filamentous molds (NDMs) have increased in recent decades, and species of the *Fusarium* genus have already been recognized as NDM pathogens of nail infections [4,5]. *Fusarium* spp. are saprophytic and of environmental origin, and some are recognized as important agricultural pathogens that are capable of devastating many crops [6]. From the late 20th century, *Fusarium* NDM became known as an emerging human pathogen when initially some species were found to affect immunocompromised patients, such as major cell transplant recipients and individuals with leukemia [7,8,9]. In these cases, *Fusarium* spp. can cause a fatal invasive fusariosis. In recent decades, infections caused by *Fusarium* spp. have also been increasing in immunocompetent patients, appearing mainly as cutaneous and localized clinical manifestations [10,11,12]. Each day, new cases of fusarial OM are added to the literature, and *Fusarium* spp. are reported as the causative agent in the majority of those caused by NDMs [13,14,15,16,17]. In immunocompetent individuals, rates of OM caused by *Fusarium* spp. have been reported to be up to 12% of all OM cases [15]. Importantly, OM can serve as a gateway for fungi to cause an invasive infection [18,19]. 

The treatment of OM demands strict adherence and requires several months to reach a clinical and mycological cure. In drug therapy, systemic antifungals are considered the gold standard, but they still have limitations [20]. Topical medications are considered as a preferable alternative in the development of OM treatments and could improve the quality of care for the patient [21]. As topical treatments act in situ, and thus have fewer adverse effects and drug interactions, they are often considered safer, which is particularly important to certain groups, such as children, the elderly and polypharmacy users. However, an important challenge in OM treatment is the formation of fungal biofilms on the nail, which limits the permeation of the antifungal [20]. 

Propolis is a natural product that has shown promise in the treatment of clinically important infectious diseases [22]. It is a gum-resinous product of the bees’ work which has various biological functions, including antifungal, antibiofilm and anti-inflammatory [23]. These properties act synergistically and are attributed to the complex constitution of the propolis which is present in its hydroalcoholic extract (propolis extract, PE), highlighting phenolic compounds as the main constituents with biological activity. Many studies have tested PE against planktonic cells [24,25,26,27,28], as in fungal biofilms formed in vitro [29,30]. In addition, PE has already been used in the successful treatment of patients with fusarial OM [28,30]. However, some features of PE, such as the strong odor and yellowing of the nail plate, have been frequently reported by patients as limiting factors affecting adherence to treatment.

The current study was based on an OM case affecting an immunocompetent individual, whose etiological agent was initially identified as belonging to the *Fusarium solani* species complex (FSSC) and afterwards definitively identified by molecular approaches as *Neocosmospora keratoplastica*. We proposed an exclusively topical treatment with a gel containing 30% propolis (PG), which showed satisfactory results. Through, a series of experiments, we aimed to deepen knowledge of fusarial OM and the rationale for the success of this new treatment. In addition, the antifungal activity of this gel and of the PE used in its production was evaluated, and the minimum inhibitory concentration and minimum fungal concentration were assessed, both in planktonic cells and biofilm. Furthermore, we characterized the biofilm formed by *N. keratoplastica* on healthy human nails, with determination of metabolic activity, total biomass, colony-forming units and scanning electron microscopy. These experiments aimed to deepen the knowledge of fusarial OM and the justification for the success of this new treatment.

## 2. Materials and Methods

### 2.1. Fungus Origin and Patient’s History 

The study was carried out based on a case of a 52-year-old Brazilian woman with suspected OM. Her medical history indicated no known underlying immunosuppressive diseases. She reported social discomfort when wearing open shoes that expose her nails. According to her report, the infection had started 15 years ago, without drug treatment at any time. According Lipner, 2021 guidelines [31], the clinical analysis of the nail plates showed moderated lesions on the big toenail of the left foot with darkened spots. The patient reported itching and local pain sporadically. Scrapings from the lesions were collected and processed at the Teaching and Research Laboratory in Clinical Analysis (LEPAC), Division of Medical Mycology at the State University of Maringá. Confirmatory laboratory tests were performed, and a direct mycological exam (DME) was performed after clarification with 20% potassium hydroxide (KOH) with Evan’s blue (3:1). The nail scrapings were also cultured on Sabouraud dextrose agar (SDA, KASVI, São José dos Pinhais, Brazil) and Mycosel agar (BD, BBL™, Les Merciers, Rhone-Alpes, France), with three points of inoculation in three tubes with each kind of culture medium. The cultures were incubated at 25 °C, and the fungal growth was assessed daily. The grown fungus was identified initially by its macromorphology and micromorphology characteristics. For molecular identification, polymerase chain reaction (PCR) amplifying the fungal DNA with internal transcribed spacers (ITSs) was performed according to O’Donnell et al. (2010) [32]. Sequences obtained were blasted in NCBI Blast (http://www.ncbi.nlm.nih.gov/BLAST, accessed on 8 February 2022), edited in Bioedit, and the phylogenetic tree was generated with MEGA11. The clinical isolation was deposited in the Microbial Collections of Paraná Network-TAXonline at the Federal University of Paraná under code CMRP5488. 

This study was made according the resolution 196/1996 from the CNS-MS (National Council of Health), under the supervision of the Standing Committee on Ethics in Research Involving Human Beings, with registration number 44.896515.5.0000.0104. The patient provided written and signed consent for publication of the images in Figure 1.

### 2.2. Propolis Formulations 

#### 2.2.1. Propolis Extract (PE)

The PE used in this study was purchased from Florien Fitoativos Ltda.™ (Maringá, Brazil) accompanied by its analysis certification, which includes the approval of pre-analytical tests carried out for product validation. The PE was hydroalcoholic and displayed a relative density (25 °C) of 0.9080 g/mL, pH 5.0, ratio of drug raw material of 20% (*w*/*w*) and total alcohol content of 63°GL. The total phenol content (TPC) of the PE was determined in the Laboratory of Research and Development of Drug Delivery Systems of the State University of Maringá through the Folin–Ciocalteau method [33]. In 25.0 mL volumetric flasks, 10 mL of purified water was added along with 10.0 μL of PE. Then, 1.0 mL of phosphomolybdotungstic reagent R (Folin–Ciocalteau) was added, and the volume was made up with 14.06% (m/V) aqueous sodium carbonate (Na_2_CO_3_) solution. After 15 min, the solution was analyzed using a Shimadzu UV-1800 spectrophotometer (Kyoto, Japan) at a wavelength of 760 nm.

#### 2.2.2. Propolis Gel (PG)

A gel containing 30% (*w*/*w*) of PE was developed in partnership with a local compounding pharmacy (Ervanário, Maringá, Brazil). The base gel was composed of EDTA, glycerin, methylparaben, Phenonip™, Carbopol 940 and water. The TPC present in the gel was estimated at 0.09% (*w*/*w*). 

#### 2.2.3. Patient Treatment

An exclusively topical treatment was proposed, with this gel containing propolis, applied once a day in a dosage determined from the bottle dispenser (approximately 200 μg). Some recommendations about disinfection of the possible fungal reservoirs were made to the patient, including washing shoes and socks and post-use of Lysoform™ spray [34]. The patient was monitored every month to evaluate the treatment evolution; in all visits, researchers asked about her experience, including difficulties or side-effects. 

### 2.3. Characterization of Fungal Growth and Biofilm Formation of the Clinical Isolate in an Ex Vivo Model

#### 2.3.1. Preparation of Inoculum 

The clinical isolate was grown on SDA for four days at 25 °C. Gently, the colonies were scraped, harvested in 0.1 M phosphate-buffered saline at pH 7.5 (PBS) and vortexed vigorously. The suspension was filtered in a syringe with sterile glass wool to obtain a pure conidial suspension. The inoculum was adjusted to the required concentration in each experiment by counting on a Neubauer chamber (Cral, Cotia, Brazil).

#### 2.3.2. Pour Plate

Healthy nail fragments from adult volunteers were selected and autoclaved at 121 °C for 20 min. A 500 μL fungal inoculum (1 × 10^7^ conidia/mL) was incorporated at a ratio of 1:100 into an amino acid-free yeast nitrogen base agar (YNB agar, BD, Difco™, Washington, DC, USA), without addition of carbohydrates, while it was liquefied in a Petri dish, 90 mm of diameter (Olen, São José dos Pinhais, Brazil). The nail fragments were then planted into the agar after solidification. Cultures were incubated at 25 °C, and the macroscopic and microscopic aspects of the colonies were evaluated after 48 h.

#### 2.3.3. Fungal Growth on the Nail Surface

Sterilized nail fragments were placed on a clean slide, ventral surface facing up, and a 3 μL fungal inoculum at 1 × 10^7^ conidia/mL was carefully distributed over each nail fragment and incubated for up to seven days at 35 °C in a humid chamber. The growth was evaluated at 24, 96 and 168 h. Prints of some infected nail fragments were made on SDA after 96 h of growth. Other fragments of infected human nails were washed twice with PBS and fixed in 0.1M cacodylate buffer solution plus 2.5% glutaraldehyde. After that, gradual dehydration was carried out with immersion for 15 min at each concentration of alcohol (50%, 70%, 80%, 90%, 95% and 100%), then critical point (Bal-TEC CPD 030) was utilized to replace the alcohol with carbon dioxide inside the cells, followed by metallization (Bal-TEC—SCD 050) and visualization of the samples with a scanning electron microscope (Shimadzu—SS-550), according to Veiga (2022) [35]. Regarding the quantitative analysis, the number of cultivable cells was determined by transferring three nail fragments to a microtube containing PBS, vortexing for 5 min, and then sonicating (Sonic Dismembrator Ultrasonic Processor, Fisher Scientific, Waltham, MA, USA) at 35% for 10 s. The resultant liquid was serially diluted in PBS, spread on SDA using the drip technique and the plates were subsequently incubated for 48 h at 25 °C to determine the colony-forming units (CFU) per mL [35]. Moreover, at each time-point, three nail fragments were transferred to microtubes containing 2,3-bis (2-metoxi-4-nitro-5-sulfofenil)-2H-tetrazólio-5-carboxanilida solution (XTT; Sigma-Aldrich, St. Louis, MO, USA), incubated for 3 h at 25 °C, and then read at 492nm in an absorbance microplate spectrophotometer (SpectraMax^®^ Plus384, Molecular Devices, Shanghai, China) in order to determine the mitochondrial metabolic activity.

### 2.4. In Vitro Effects of Propolis on the Isolated Fungus

#### 2.4.1. Antifungal Activity

Minimal inhibitory concentration (MIC) was determined using the broth microdilution method, according to Galletti et al. (2015) based on CLSI methods with some modifications for natural products [36,37]. MIC was defined as the lowest concentration of TPC capable of inhibiting fungal growth. In a 96-well plate (Jet-BioFil, Guangzhou, China), 100 μL RPMI 1640 medium (Sigma-Aldrich, Sao Paulo, Brazil) was aliquoted into each well, 100 μL of PE or 100 μg of PG was added to the first wells and homogenized. Then, the compounds were serially diluted (PE—2.93 to 1,500 μg/mL TPC and PG—0.87 to 450 μg/mL TPC) throughout the plate. Finally, 100 μL of fungal inoculum (at 1 × 10^4^ conidia/mL) was added, and the plate was incubated at 25 °C for 24 h. After this time, 30 μL of resazurin was added to each well and incubated again for another 24 h under the same conditions as above [38]. RPMI plus the inoculum and RPMI only were used in some wells as growth and negative controls, respectively. The minimum fungicidal concentration (MFC), that is the lowest concentration of TPC capable of preventing fungal growth, was estimated by transferring 10 μL from each well of the MIC plate to a plate containing SDA and incubating at 25 °C for 48 h then assessing growth visually.

#### 2.4.2. Antibiofilm Activity 

Minimal biofilm inhibitory concentration (MBIC) was determined using the microdilution method, based on the above technique with adaptations [29]. Biofilms were generated in the wells of a 96-well plate using 100 μL of fungal inoculum at a concentration of 1 × 10^6^ conidia/mL in RPMI and incubating the plate for 24 and 48 h at 25 °C. After biofilm formation, washes were carried out with PBS to remove the non-adherent cells, then 100 μL RPMI was added to all wells, followed by the addition of 100 μL of PE or 100 μg of PG to the first wells, which were then serially diluted. The plate was then incubated at 25 °C for 24 h. For the reading, 30 μL of resazurin was added to the wells, and the plate was incubated again for another 24h [38]. The minimal biofilm eradication concentration (MBEC) was defined as the lowest concentration of TPC capable of preventing the fungal growth and was estimated using the same method as that used for MFC determination. 

#### 2.4.3. Impact of PE Exposure to the Biofilm 

Biofilms were produced with a 1 × 10^7^ conidia/mL inoculum in 24-well plates (KASVI, São José dos Pinhais, Brazil at three different times: 2, 96 and 168 h. After the defined formation period, the wells were washed with 500 μL PBS to remove non-adherent cells, and then 300 μL of PE (375 µg/mL of TPC) was added, and the plate was incubated at 25 °C for 24 h. After this, wells were washed again with 500 µL PBS. Negative controls with RPMI medium only and positive growth controls not exposed to PE were also included. The evaluations were performed according to methodologies previously described [35,36]. SEM analyses were performed following the conditions described above for PE and PG in mature biofilms (168 h of growth).

#### 2.4.4. Statistical Analysis

The software GraphPad Prism 5 (OSB, Northampton, USA) was used for statistical analysis. Data were statistically evaluated with one- and two-way analysis of variance (ANOVA), followed by the Bonferroni multiple comparison test. Data with a non-normal distribution were expressed as the mean ± standard deviation (SD). For each assay, at least three independent experiments were performed and on three different days. Values of *p* < 0.05 were considered statistically significant.

## 3. Results

Figure 1 shows the clinical and microbial features before and after the onychomycosis treatment with propolis gel for 12 months. DME showed a moderate presence of fungal structures. In the culture, cream-brown colonies, with opaque and slightly cottony aerial mycelium, grew on SDA but not on Mycosel agar. The micromorphological aspects of these colonies showed hyaline hyphae as well as curved and spindle-shaped macroconidia. The fungus was molecularly identified by the ITS gene as *Neocosmospora keratoplastica*.

Figure 2 displays the ability of this isolate to grow on the human nail, without the addition of other nutrients. Macroscopic growth on the nail fragments both in the environment (Figure 2A) and soaked in poor agar (only salts) without nutritional sources (Figure 2B) are easily viewed. In addition, it was possible to prove its perfect development as the colonial characteristics exhibited were reproduced later in a complete culture medium (Figure 2C).

The development of *N. keratoplastica* in biofilm, using the nail as the only nutritional source, can be observed in Figure 3. The image captured by SEM shows that in 24 h a large quantity of isolated conidia adhered to the nail surface, also revealing hyphae at the beginning of development (Figure 3A). At 96 h, attention was drawn to the constant presence of fungal structures, similar to banana bunches, formed by conglomerates of overlapping conidia (detail), in addition to the development of a network of hyphae (Figure 3B). At 168 h, a dense, well-organized biofilm was observed, with a predominance of hyphae (Figure 3C). At this time, the conidia that emerged in the biofilm exhibited typical morphology of a fungal biofilm architecture, such as the tunnels formed on the depressions of the nail plate (detail).

The formed biofilm in nails showed a significant increase in the number of colony-forming units (CFU) recovered from the biofilm at 96 h (*p* < 0.0001) (Figure 4A). Meanwhile, the metabolic activity decreased significantly (*p* = 0.0013) (Figure 4B).

The TPC test of the PE showed the 0.30% of polyphenols, the in vitro susceptibility of which for this clinical isolate is presented in Table 1. MIC and MBIC were generally identical to the respective MFC and MBEC for both PE (375 µg/mL of TPC) and PG (450 µg/mL of TPC). Comparing the antifungal activity of propolis on planktonic and sessile cells, it was evidenced that very close or equal concentrations of each product were sufficient to inhibit or prevent the growth of the fungus.

The exposure to propolis also had an important impact on the fungal cells when they were organized in the biofilm form, mainly causing a decrease in the number of CFU (Figure 5A). Cells adhered for 2 h and preformed biofilm (96 and 168 h) when exposed to PE (375 µg/mL of TPC) showed a significant decrease in CFU/mL compared to the control, not exposed to propolis but subjected to the same conditions. There was also a decrease in metabolic activity at 96 and 168 h in comparison to the untreated control (Figure 5B). On the contrary, at 2 h (adherence period), a value significantly lower than the control was observed (*p* < 0.001). Furthermore, the total biomass increased after PE exposure at each of the time-points (Figure 5C).

Images captured by SEM illustrate the impact of PE and PG on the mature biofilm (168 h) produced for *N. keratoplastica*. Biofilm exposed to PE, despite maintaining the basic architecture, showed a drastic reduction in the extracellular matrix (ECM) and number of conidia (Figure 6B). In the biofilm exposed to PG, there was a visible decrease in hyphae and conidia, accompanied by a disorganization of the ECM (Figure 6C). Ruptures were also visualized as lesions throughout the preparation.

## 4. Discussion

FSSC is a heterogeneous complex that clearly varies regarding host preferences, from plants to humans. Species of fusarioid fungi are closely related and hardly distinguishable morphologically, so for correct identification, molecular methods are necessary. Recently, a taxonomic remodeling was proposed for the organization of fusarioid fungi [39]. The clinical isolate of this study was presumptively identified based on typical macro- and micromorphology as a species of the FSSC, but according to the new nomenclature it was molecularly identified as *Neocosmospora keratoplastica*. This fungus corresponds to the ancient *Fusarium keratoplasticum*, which has already been isolated from OM [40].

There are several cases of OM caused by *Fusarium* spp. described in the literature [13,16,17,41,42,43,44]. In South America, this incidence is higher than in other parts of the world [45]. Curiously, in our region (Southern Brazil), these rates range from 7.5% to 12.4% of all OM cases [14,15,36]. It can probably be attributed to the region being heavily agricultural and to the adaptation of the fungus, under pressure from pesticides, to human tissues. 

The diagnosis of OM, with identification of the etiologic agent, is extremely important for the determination of an effective drug therapy. In this case, in just seven days it was possible to obtain a presumptive mycological diagnosis, due to the fast growth of the fungus present in the biological sample. Due to environmental origin, *Fusarium* spp. was considered for a long time as a contaminant in the laboratorial routine, but it is currently recognized as a primary pathogen [14,46]. The clinical isolate used in the current study was obtained from a 52-year-old Brazilian woman, with suspected onychomycosis. Her medical history had no known immunosuppressive underlying disease. According to her report, the infection had started 15 years ago, without drug treatment at any time. After clinical analysis of the nail plates, lesions were found on the left big toenail with darkened spots. The patient reported itching and local pain sporadically. Fungal structures (Figure 1C) compatible with those previously reported [14] were revealed in the DME. We assumed that the isolated fungus was the agent of onychomycosis and not a contaminant.

Nevertheless, the etiopathogenesis of OM by *N. keratoplastica* is still unclear, so we initially carried out a laboratory investigation regarding criteria that could prove its etiology as a primary pathogen of nail infection. We proved the ability of the clinical isolate to develop on the nail as only a nutritional substrate (Figure 2), similarly to that described for *F. oxysporum* [35]. In addition, Figure 3 shows the agent’s ability to form biofilms on the healthy nail. SEM images reveal typical structures; in 96 h, hyphae were organized in a network with numerous conidia clusters, and in 168 h, the presence of a mature structure and dense biofilm was evident. The biofilm characterization was reinforced by the number of CFU/mL and metabolic activity (Figure 4). The metabolism in the adaptive phase (24 h) appeared to be more intense, presumably due to the *N. keratoplastica* using the energy reserve brought over from its previous cultivation. On the other hand, in 96 h there was a decrease in metabolic activity, suggesting that at this time the fungus, in biofilm, survived exclusively on nutrients extracted from the nail, since there was no other nutritional source.

OM etiopathogenesis is closely related to the formation of biofilms by its etiological agents, guaranteeing protection, nutrition and resistance to treatment, thus being difficult to eradicate [47]. Initially, this association was consolidated for the dermatophyte fungi “dermatophytoma” [48,49], however, recent studies have provided strong evidence that fusarial OM is also due to a biofilm formed by an NDM [20,35,47]. Propolis is a promising natural compound for use as a treatment as it is safe for human use and has low cost and benefits other than antifungal properties, such as anti-inflammatory, antioxidant and healing [23,50]. Propolis works by curing the infection and recovering the adjacent tissues, which is particularly attractive in cases where the OM is accompanied by inflammatory processes that result in clinical manifestations such as pain, swelling, paronychia and itching. Propolis has already been used topically on infected nails in the form of an extract [5,28,30]. However, some limitations such as the yellowing of the nails and the strong odor during prolonged treatment need to be improved. In the present case, we proposed a topical preparation of propolis, with a pharmaceutical formulation in the form of a colorless gel. The active principle was defined based on the excellent results in studies involving PE that proved its efficient fungicidal action [27,29,51]. In the current study, initially, the MIC and MFC for planktonic cells and the MBEC and MBIC for in vitro formed biofilm were determined for both the raw material PE and the PG specially formulated for this patient. The results were encouraging (Table 1), strongly suggesting antifungal and antibiofilm activity, since both PE and PG were efficient in inhibiting and preventing the growth of *N. keratoplastica*. The determination of MIC and MFC on planktonic cells is widely used as an indicator of the antifungal action of a drug in vitro. Despite this, the parameter of MBEC is supposedly more consistent with the conditions encountered during OM, being more suitable for predicting the effectiveness of antifungals in vivo [52], as it indicates propolis has acted on the preformed biofilm that is naturally found in an OM.

Exposure of experimentally formed *N. keratoplastica* biofilms to PE significantly impacted the number of fungal cells and metabolic activity while there was an increase in total biomass (Figure 5). This data set appears contradictory; however, the images obtained by SEM (Figure 6) suggest that exposure to PE seems to induce a vegetative mycelium production, which could be responsible for the increase in total biomass. This hypothesis is reinforced by the statistically significant decrease in CFU compared to the untreated control. Despite maintaining the basic architecture of a mature biofilm, PE provoked an important reduction in the ECM and the reproductive cells (conidia). Meanwhile, PG also decreased hyphae and conidia, being apparently less effective on the ECM; however, PG could provoke ruptures by disorganizing it. These data confirm the efficient antifungal activity of the PE, with good results including on chronic nail infections with strong evidence of biofilm. Recently, we reported a 50-year case of OM caused by *T. rubrum*, which was treated efficiently with PE [28].

According to our in vitro results, PG was less efficient than PE; however, the treatment of the patient with PG was successful, with complete cure in 12 months, providing the same good antifungal result is already obtained with PE. Additionally, the aesthetic issues encountered by the PE were greatly improved by the PG, increasing the good acceptance by the patient; this is crucial since adherence to treatment is necessary to achieve healing. We were very attentive to any suggestions in order to adapt the treatment in the best possible way. Based on the patient reports, we can state that the gel formulation offers conditions for better acceptance regarding comfort and commodity. PG is easy to apply and directly acts on the infected site, overcoming the disadvantages of PE such as the yellowish aspect that remains on the nails and the strong odor.

Finally, we would like to note some limitations of the study, such as the lack of an antifungal molecule known used as a control. The confirmation of the antibiofilm action of the propolis extract is still preliminary, as studies involving the kinetics and growth curves of the fungus with exposure to propolis are still ongoing in our laboratory.

## 5. Conclusions

Here, we demonstrate the ability of *N. keratoplastica* to be an excellent biofilm former, especially on the nail, without any external nutrient source. Additionally, we present a case of chronic OM treated exclusively with a gel containing 30% propolis. We reinforced the excellent antifungal activity of propolis and showed the efficiency of PG, after providing complete cure in 12 months, both from clinical and mycological aspects, raising the possibility of its use in the treatment of OM.

## Figures and Tables

**Figure 1 jof-08-01216-f001:**
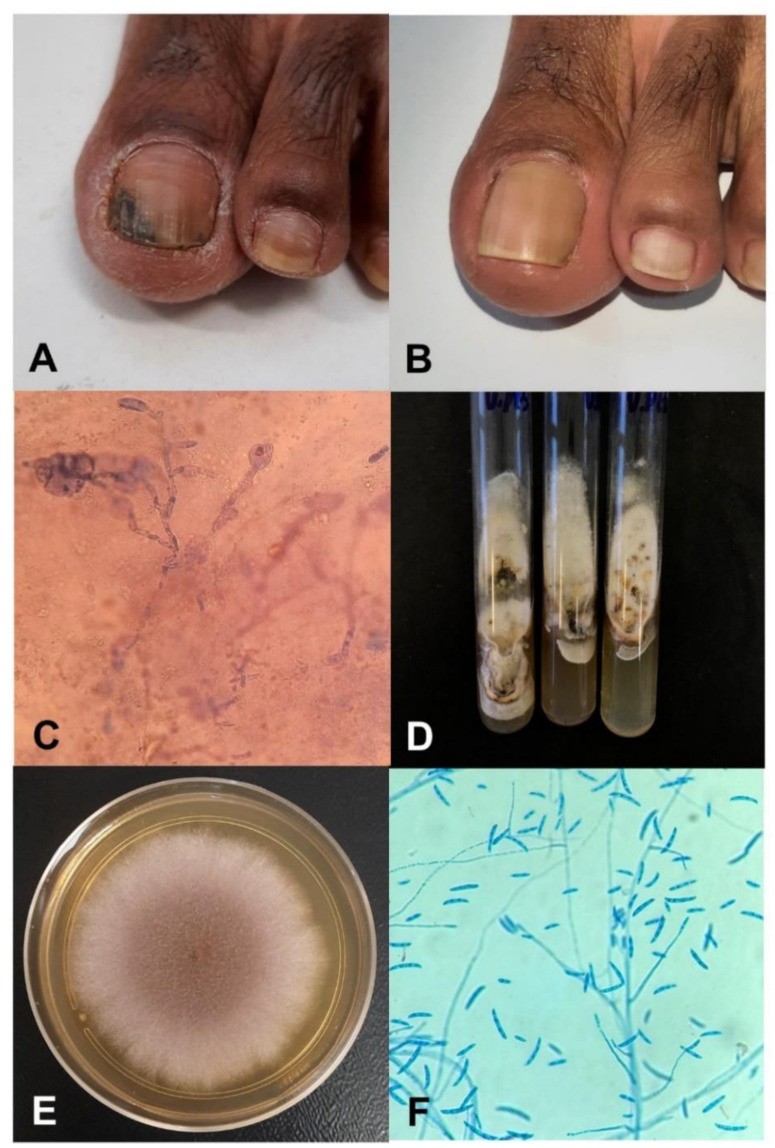
Clinical and laboratory characteristics of a case of onychomycosis due to *Neocosmospora keratoplastica* topically treated with propolis gel for 12 months. (**A**) Clinical nail aspects before treatment; (**B**) after treatment; (**C**) fungal structures observed with direct microscopy after clarification with KOH plus Evans. (**D**) development in Sabouraud dextrose agar after four days of incubation at 25 °C; (**E**) macro- and (**F**) micromorphological aspects of the grown colonies.

**Figure 2 jof-08-01216-f002:**
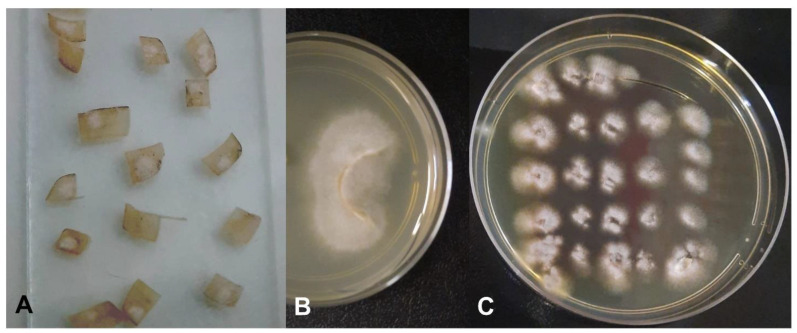
Ability of *Neocosmospora keratoplastica* to grow on the human nail using it as the only substrate. (**A**) Macroscopic growth on the nail surface in four days at 25 °C in a moist chamber; (**B**) the pour plate technique with the inoculum incorporated into the YNB agar base, without adding of carbon sources, proving fungal growth restricted to the health nail; (**C**) Print of infected nail fragments on Sabouraud dextrose agar complete medium.

**Figure 3 jof-08-01216-f003:**
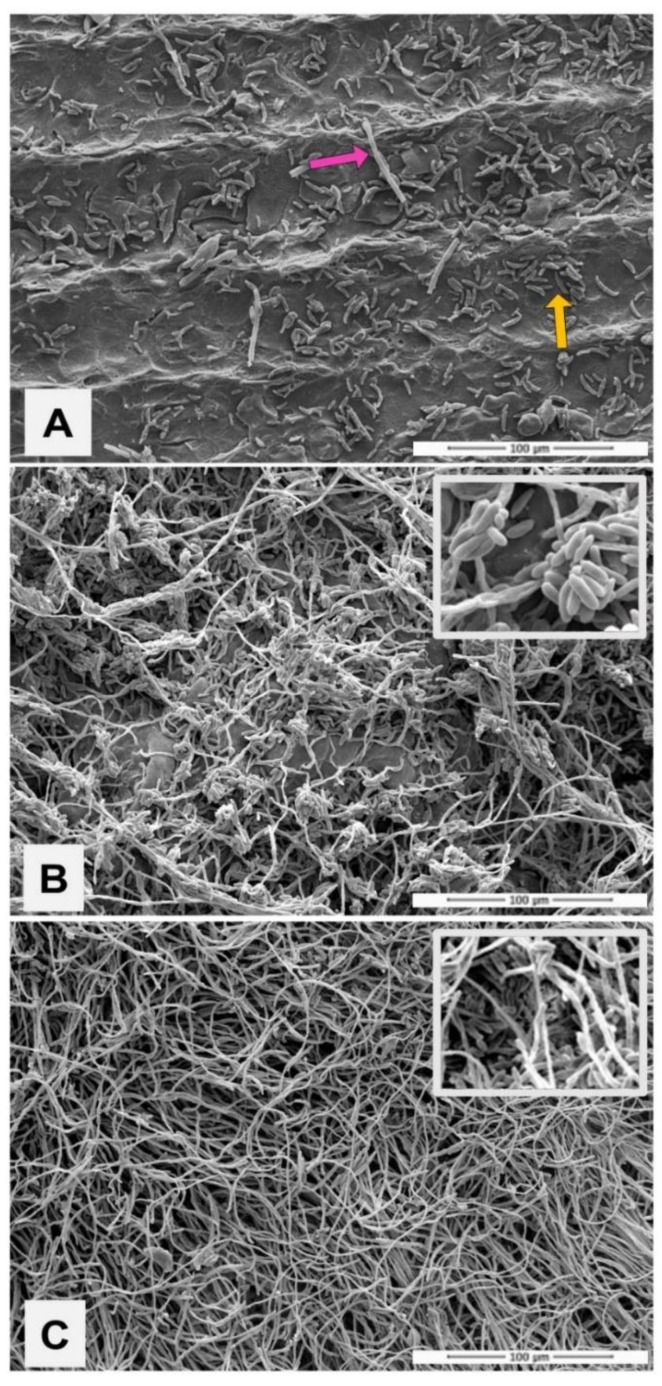
Scanning electron microscopy microphotographs showing the architecture of the biofilm produced by *Neocosmospora keratoplastica* using healthy and sterilized human nails as a unique nutritional source. (**A**) Within 24 h, a large number of conidia (yellow arrow) adhered to the entire surface of the nail, with the presence of developing hyphae (pink arrow); (**B**) after 96 h, the formation of hyphae is seen along the entire length of the nail and a large quantity of conidia in the shape of bunches (detail); (**C**) in 168 h, the biofilm can be seen already formed and organized, showing a dense tangle of hyphae and the presence of few conidia. SEM images at 1000× magnification.

**Figure 4 jof-08-01216-f004:**
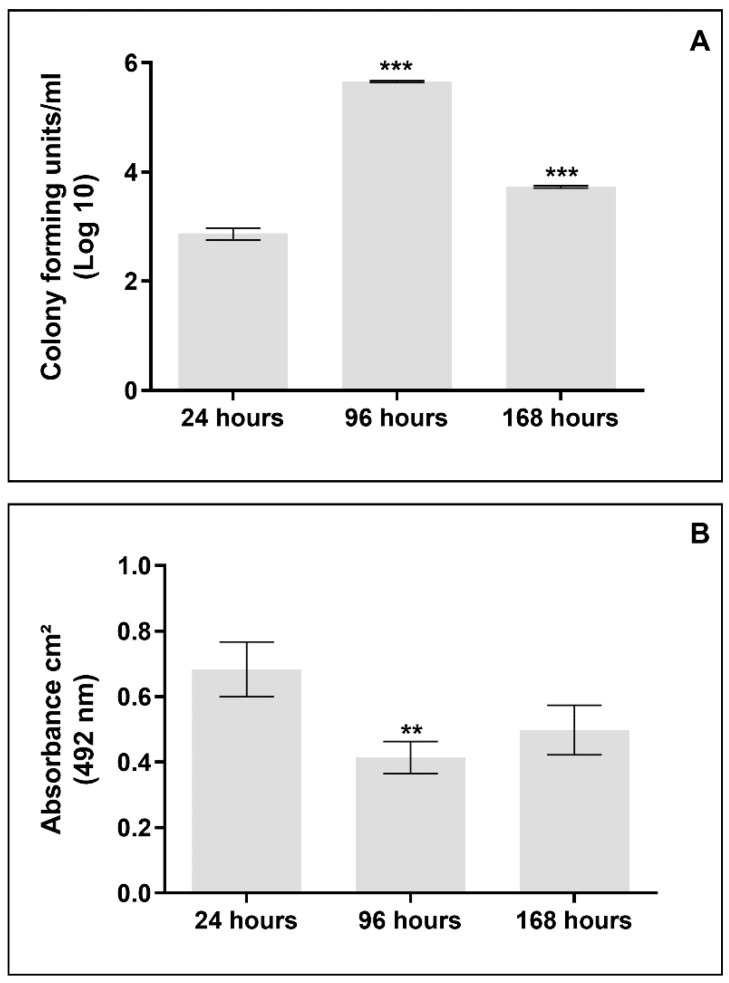
Viability of *Neocosmospora keratoplastica* cells organized in biofilm produced on sterilized healthy human nails. (**A**) The number of CFU recovered from biofilm increased significantly (***) between 24 and 96 h (*p* < 0.0001) and decreased (***) between 96 and 168 h (*p* < 0.0001); (**B**) metabolic activity revealed by XTT indicated that there was a statistically significant decrease (**) between 24 and 96 h (*p* = 0.0013) with a slight increase (not significant) at 168 h.

**Figure 5 jof-08-01216-f005:**
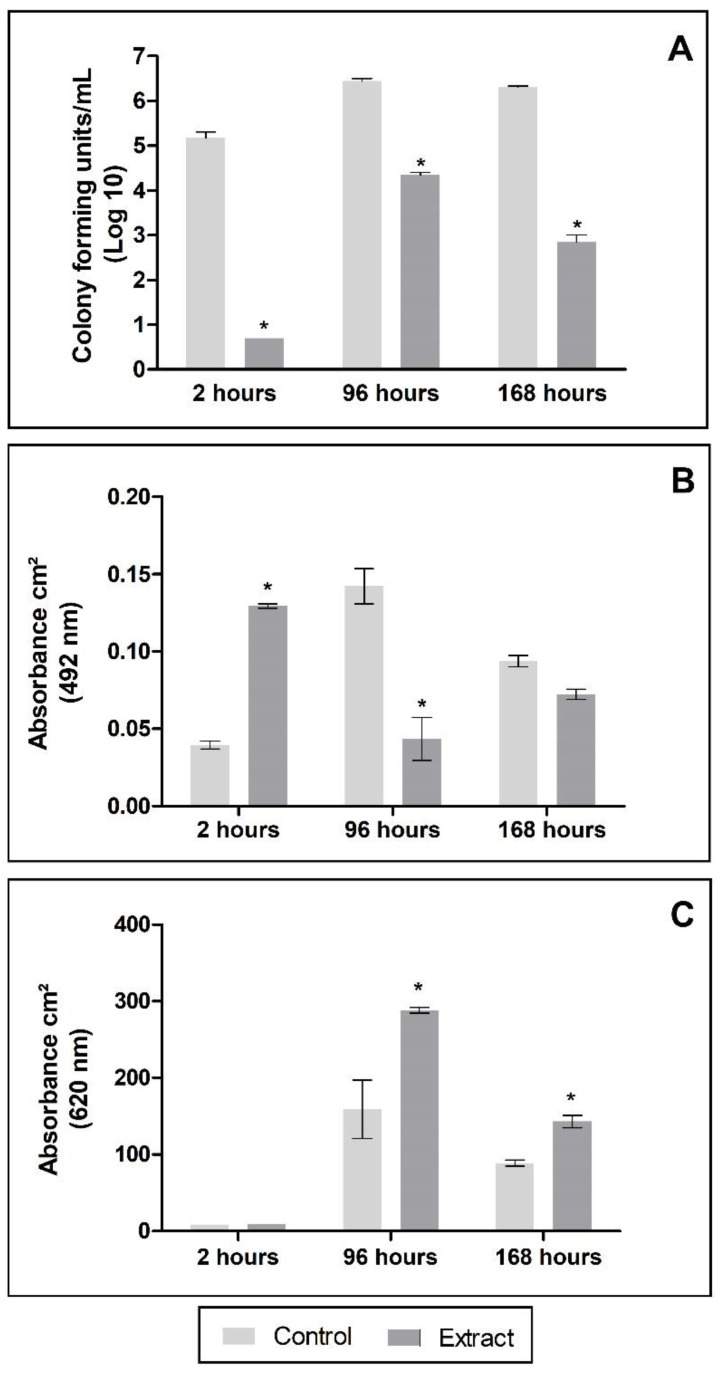
Effects of propolis extract on *Neocosmospora keratoplastica* biofilm. (**A**) Significant inhibition (*) of the number of cells during the formation of the biofilm and the already formed biofilm (*p* < 0.001); (**B**) metabolic activity reduction is statistically significant (*) in biofilms of 96 h (*p* < 0.001); (**C**) significant increase (*) in total biomass in 96 h (*p* < 0.001); 168h (*p* < 0.05).

**Figure 6 jof-08-01216-f006:**
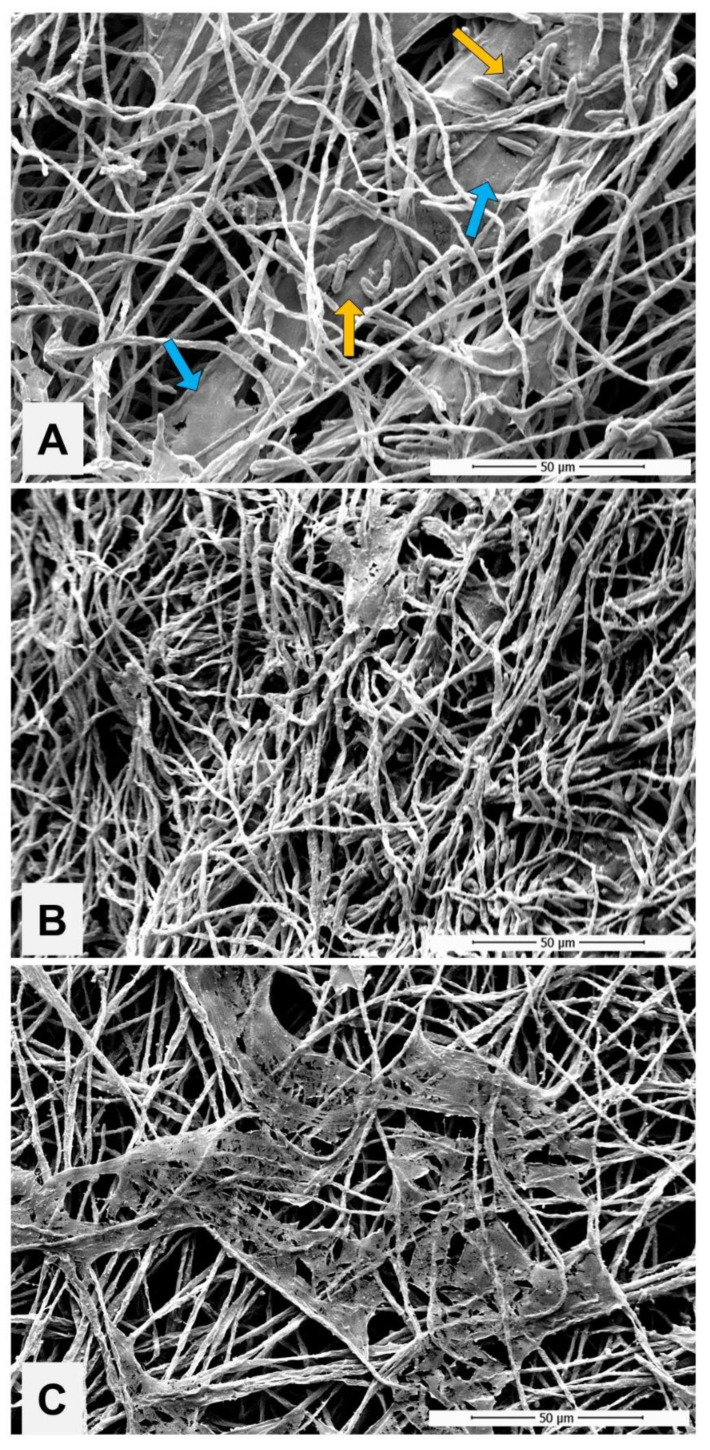
Seven-day biofilm produced by *Neocosmospora keratoplastica* on polystyrene plate. (**A**) Morphology of the control biofilm presenting a dense tangle of hyphae, conidia (yellow arrow) and the presence of intense extracellular matrix (blue arrow); (**B**) biofilm exposed to propolis extract showed a drastic reduction in matrix and conidia but maintained its architecture; (**C**) biofilm exposed to propolis gel evidencing a reduction in hyphae and conidia and a torn extracellular matrix. SEM images at 2000× magnification.

**Table 1 jof-08-01216-t001:** Evaluation of the minimum inhibitory concentration (MIC), minimum biofilm inhibitory concentration (MBIC), minimum fungicidal concentration (MFC) and minimum biofilm eradication concentration (MBEC) of propolis extract and propolis gel against planktonic cells and biofilms (24 and 48 h) of *Neocosmospora keratoplastica* isolated from onychomycosis.

	MIC/MBIC	MFC/MBEC
Extract	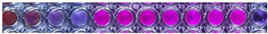	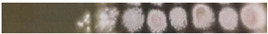
Gel	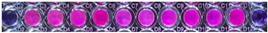	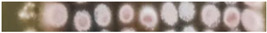
	Extract *	Gel *	Extract *	Gel *
Planktonic	187.50	450	375	450
24 h biofilm	375	450	375	450
48 h biofilm	375	450	375	450

* Data expressed in µg of total polyphenols per mL.

## Data Availability

Not applicable.

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
