# Peer review of "Efficacy of Propolis Gel on Mature Biofilm Formed by Neocosmospora keratoplastica Isolated from Onychomycosis"

_jof, 2022, doi:10.3390/jof8111216_

Round 1

Reviewer 1 Report

1. Materials and Method (Fungus origin): Please put details (re volume, concentration, complete PCR steps (ref 31) in the section so that the readers do not have to go to other references to replicate the method. 

2. please put details re ref 32.

3. Lines 122-135: please put detailed methodology including sources of materials.

4. Lines 143-144. Please put detailed method and parameters re SEM. ref 34.

5. Lines 151-152: Please put the  brand and type of the spectrophotometer.

6. Lines 319-320: Not too sure about the term "consensual" here. 

 Overall comment: This study reinforces the potential use of propolis as an anti-fungal treatment. However, the methodology section requires more details. 

Author Response

RESPONSES TO REVIEWER 

Title: Efficacy of propolis gel on mature biofilm formed by Neocosmospora keratoplastica isolated from onychomycosis

Reference number:  jof-2005668

Ms. Yasmine Wang

Assistant Editor Journal of fungi

We would like to thank the reviewers for careful and thorough reading of this manuscript and for the thoughtful comments and constructive suggestions, which help to improve the quality of this manuscript. We have studied these comments carefully and have made corresponding corrections that we hope will meet with your approval. 

All changes are evidenced with the “Track Changes” function like solicited.

Our response follows (the reviewer’s comments are in italics). 

Comments and suggestions from the reviewers to authors:

Reviewer 1

  1. Materials and Method (Fungus origin): Please put details (re volume, concentration, complete PCR steps (ref 31) in the section so that the readers do not have to go to other references to replicate the method. 

Author's response: It was not possible to answer this question because the molecular identification was performed by the Microbiological Collections Of Paraná Network (CMRP/Taxonline), Paraná, Brazil where it was deposited under the number CMRP5488. The CMRP uses validated and confidential  protocols, which are not made available to the depositors.

  1. please put details re ref 32.

Author's response: Suggestion accepted. We added the information on the methods on page 3, lines 134-141 and we mentioned the result on page 10, lines 325, It is worth mentioning that reference 32 is now 33.

  1. Lines 122-135: please put detailed methodology including sources of materials.

Author's response: The authors inform that the methodology used according to Veiga, 2022, is already detailed in the manuscript. The origins of the culture media were informed in previous items of this methodology, the PBS buffer was formulated in-house and the origin of the Neubauer Chamber and the Petri dish were added as requested.

  1. Lines 143-144. Please put detailed method and parameters re SEM. ref 34.

Author's response: The reviewer's suggestion was accepted, the information below was entered on page 4, lines 183-189.

“In each of the three evaluation periods, infected nails were washed twice with PBS, and fixed in 0.1M cacodylate buffer solution plus 2.5% glutaraldehyde. After there, a gradual dehydration with immersion for 15 min at each concentration of alcohol (50%, 70%, 80%, 90%, 95% and 100%), then they are submitted to the critical point (Bal-TEC CPD 030) to replace the alcohol with carbon dioxide inside the cells, metallization (Bal-TEC - SCD 050) and visualization of the samples in the scanning electron microscope (Shimadzu - SS-550)”.

  1. Lines 151-152: Please put the  brand and type of the spectrophotometer.

Author's response: Suggestion accepted. 

  1. Lines 319-320: Not too sure about the term "consensual" here. 

Author's response: Suggestion accepted, the word “consensual “ has been excluded. 

Overall comment: This study reinforces the potential use of propolis as an anti-fungal treatment. However, the methodology section requires more details. 

Author's response: That is ok, authors thank for relevant suggestions, which all of them were accepted. 

Reviewer 2 Report

Dear authors,

I read your manuscript concerning the ex vivo, in vivo and in vitro study of propolis gel and extract in the treatment of biofilm and onychomycosis. I really appreciated the translational approach of your work, both for the clinical and the basic aspects.

1)      Line 46, correct

2)      Line 49-50, I suggest the authors reevaluate the guidelines. The American ones distinguish therapy based on age and nail involvement on the OSI index and timeline of infections. In contrast, the European guidelines are based on the isolated microorganism and on the OSI index, setting specific treatment algorithms. In the same way, in the guidelines and in clinical practice, pulse therapy can be performed, which increases patients' adherence to therapy. Read and cite:

-          Lipner SR, Joseph WS, Vlahovic TC, et al. Therapeutic Recommendations for the Treatment of Toenail Onychomycosis in the US. J Drugs Dermatol. 2021;20(10):1076-1084. doi:10.36849/JDD.6291

3)      When we talk about biofilm in nails, we talk about Dermatophytoma as a clinical recognized sign and a correspondence biofilm. Read  and cite:

-          Leeyaphan C, Bunyaratavej S, Prasertworanun N, Muanprasart C, Matthapan L, Rujitharanawong C. Dermatophytoma: An under-recognized condition. Indian J Dermatol Venereol Leprol. 2016 Mar-Apr;82(2):188-9. doi: 10.4103/0378-6323.165539. PMID: 26374659.

-          https://doi.org/10.1016/j.jaad.2014.01.451

4)      Line 72, report all the experiments you performed and the aim of this article.

5)      Line 115-117. how do you prove that the resolution of onychomycosis is not due to the use of lisoformTM but the gel?

6)      Line 150, XTT full name, please

7)      Line 157-158, you must refer to EUCAST or CLSI guidelines.

8)      Line 327,328. Read and cite:

- McEwen SA, Collignon PJ. Antimicrobial Resistance: a One Health Perspective. Microbiol Spectr. 2018 Mar;6(2). doi: 10.1128/microbiolspec.ARBA-0009-2017. PMID: 29600770.

- Jacques M, Malouin F. One Health-One Biofilm. Vet Res. 2022 Jul 7;53(1):51. doi: 10.1186/s13567-022-01067-4. PMID: 35799278; PMCID: PMC9264708.

9)      Line 391,405, correct

10)  I suggest reading and cite this work about biofilm and natural compounds:

-          Pistoia ES, Cosio T, Campione E, Pica F, Volpe A, Marino D, Di Francesco P, Monari C, Fontana C, Favaro M, Zampini P, Orlandi A, Gaziano R. All-Trans Retinoic Acid Effect on Candida albicans Growth and Biofilm Formation. Journal of Fungi. 2022; 8(10):1049. https://doi.org/10.3390/jof8101049

11)  The main problem of your work is the Study limitations: we don't know with the compound in the propolis extract is more active against biofilm; furthermore, for future study, it could be brought back to the kinetics with growth curves of the fungus. The study also lacks the evaluation of an antifungal molecule known as amphotericin B, to be used as a control.

12)  I absolutely agree with the use of a gel as a formulation, both for the ease of application and for the texture

13)  The patient's medical history and previous therapy are substantially missing.

Author Response

RESPONSES TO REVIEWER 

Title: Efficacy of propolis gel on mature biofilm formed by Neocosmospora keratoplastica isolated from onychomycosis

Reference number:  jof-2005668

Ms. Yasmine Wang

Assistant Editor Journal of fungi

We would like to thank the reviewers for careful and thorough reading of this manuscript and for the thoughtful comments and constructive suggestions, which help to improve the quality of this manuscript. We have studied these comments carefully and have made corresponding corrections that we hope will meet with your approval. 

All changes are evidenced with the “Track Changes” function like solicited.

Our response follows (the reviewer’s comments are in italics). 

Comments and suggestions from the reviewer 2 to authors: 

I read your manuscript concerning the ex vivo, in vivo and in vitro study of propolis gel and extract in the treatment of biofilm and onychomycosis. I really appreciated the translational approach of your work, both for the clinical and the basic aspects.

1)      Line 46, correct

Author's response: Suggestion accepted, it was rephased. 

2)      Line 49-50, I suggest the authors reevaluate the guidelines. The American ones distinguish therapy based on age and nail involvement on the OSI index and timeline of infections. In contrast, the European guidelines are based on the isolated microorganism and on the OSI index, setting specific treatment algorithms. In the same way, in the guidelines and in clinical practice, pulse therapy can be performed, which increases patients' adherence to therapy. Read and cite:

Lipner SR, Joseph WS, Vlahovic TC, et al. Therapeutic Recommendations for the Treatment of Toenail Onychomycosis in the US. J Drugs Dermatol. 2021;20(10):1076-1084. doi:10.36849/JDD.6291

Author's response: Suggestion accepted, information and respective reference have been added, but on page 2, line 88, as it fits better in the proposal, especially after following suggestion 13 below. 

3)      When we talk about biofilm in nails, we talk about Dermatophytoma as a clinical recognized sign and a correspondence biofilm. Read  and cite:

-   Leeyaphan C, Bunyaratavej S, Prasertworanun N, Muanprasart C, Matthapan L, Rujitharanawong C. Dermatophytoma: An under-recognized condition. Indian J Dermatol Venereol Leprol. 2016 Mar-Apr;82(2):188-9. doi: 10.4103/0378-6323.165539. PMID: 26374659.

-   https://doi.org/10.1016/j.jaad.2014.01.451

Author's response: Yes, we agree that the current knowledge available so far associates biofilm in nails with dermatophytoma. However, it is possible that this concept needs to be revised soon, taking into account some recent studies (https://doi.org/10.1016/j.jaad.2016.01.008; https://doi.org/10.1016/j.micpath.2022.105640; https://doi.org/10.1111/ijd.15747). Reinforcing the above thought, our group has shown strong evidence associating OM with a biofilm by Fusarium spp. (a NDM). We inform you that in consideration of the reviewer's comment, this information and the suggested authors were added to the manuscript on page 13, lines 411-413.    

4)      Line 72, report all the experiments you performed and the aim of this article.

Author's response: Suggestion accepted.

5)    Line 115-117. how do you prove that the resolution of onychomycosis is not due to the use of lisoformTM but the gel?

Author's response: Our conclusion regarding propolis efficiency was based on several experiments contained in this study. Reinforced by the fact that the base of this gel has been tested in our laboratory showing no antifungal action (data not shown in this study).

6)      Line 150, XTT full name, please

Author's response: Suggestion accepted.

7)      Line 157-158, you must refer to EUCAST or CLSI guidelines.

Author's response: Suggestion accepted.

8)      Line 327,328. Read and cite:

- McEwen SA, Collignon PJ. Antimicrobial Resistance: a One Health Perspective. Microbiol Spectr. 2018 Mar;6(2). doi: 10.1128/microbiolspec.ARBA-0009-2017. PMID: 29600770.

- Jacques M, Malouin F. One Health-One Biofilm. Vet Res. 2022 Jul 7;53(1):51. doi: 10.1186/s13567-022-01067-4. PMID: 35799278; PMCID: PMC9264708.

Author's response: We read with attention these articles, indeed are very interesting, however we have opted not to cite them since both report the biofilm produced by bacterias and our study is specifically on antifungal properties.

9)      Line 391,405, correct

Author's response: Suggestion accepted, it was corrected.

10)  I suggest reading and cite this work about biofilm and natural compounds:

-   Pistoia ES, Cosio T, Campione E, Pica F, Volpe A, Marino D, Di Francesco P, Monari C, Fontana C, Favaro M, Zampini P, Orlandi A, Gaziano R. All-Trans Retinoic Acid Effect on Candida albicans Growth and Biofilm Formation. Journal of Fungi. 2022; 8(10):1049. https://doi.org/10.3390/jof8101049

Author's response: This article is based on yeast germination and filamentous growth as pseudo-hyphae or true hyphae in C. albicans showing they are crucial events for formation and maintenance of biofilm produced by this yeast and the antifungal activity of All-Trans Retinoic Acid molecule. Therefore, we thought it was not related with the focus of the current study, which is on Fusarium, a filamentous fungi.

11)  The main problem of your work is the Study limitations: we don't know with the compound in the propolis extract is more active against biofilm; furthermore, for future study, it could be brought back to the kinetics with growth curves of the fungus. The study also lacks the evaluation of an antifungal molecule known as amphotericin B, to be used as a control.

Author's response: These are excellent suggestions, the authors acknowledge the limitations pointed out and inform that this was included in the manuscript on page 14, lines 458-462. Regarding the use of amphotericin B as a control, we understand that it would not be relevant as our focus is on onychomycosis, where this antifungal has no indication.

12)  I absolutely agree with the use of a gel as a formulation, both for the ease of application and for the texture.

Author's response: That’s ok, thank you for your comments. 

13)  The patient's medical history and previous therapy are substantially missing.

Author's response: These information were added on page 2 and lines 83-90.

Round 2

Reviewer 2 Report

Dear Authors,

all the corrections have been made.